# Dayside magnetopause reconnection and flux transfer events: BepiColombo Earth-flyby observations

Weijie Sun[1], James A. Slavin[1], Rumi Nakamura[2], Daniel Heyner[3], Karlheinz J. Trattner[4], Johannes Z. D. Mieth[3], Jiutong Zhao[5], Qiu-Gang Zong[5], Sae Aizawa[6,7], Nicolas Andre[6], Yoshifumi Saito[8]

[1]Department of Climate and Space Sciences and Engineering, University of Michigan, Ann Arbor, MI 48109, United States
[2]Space Research Institute, Austrian Academy of Sciences, Schmiedlstraße 6, 8042 Graz, Austria
[3]Institut für Geophysik und extraterrestrische Physik, Technische Universität Braunschweig, 38106 Braunschweig, Germany
[4]Laboratory for Atmospheric and Space Physics, University of Colorado, Boulder, CO 80303, USA
[5]School of Earth and Space Sciences, Peking University, Beijing 100871, China
[6]Institut de Recherche en Astrophysique et Planétologie, CNRS-UPS-CNES, Toulouse, France
[7]Department of Geophysics, Graduate School of Science, Tohoku University, Sendai, Japan
[8]Japan Aerospace Exploration Agency, Institute of Space and Astronautical Science, Kanagawa, Japan

*Correspondence to*: Weijie Sun (wjsun@umich.edu)

**Abstract.** This study analyzes the flux transfer event (FTE)-type flux ropes and magnetic reconnection around the dayside magnetopause during BepiColombo's Earth flyby. The magnetosheath has a high plasma $\beta$ (~ 8) and the interplanetary magnetic field (IMF) has a significant radial component. Six flux ropes are identified around the magnetopause. The motion of flux ropes together with the maximum magnetic shear model suggest that the reconnection X-line possibly swipes BepiColombo near the magnetic equator due to an increase of the radial component of IMF. The flux rope with the highest flux content contains a clear coalescence signature, i.e., two smaller flux ropes merge, supporting theoretical predictions the flux contents of flux ropes can grow through coalescence. The coalescence of the two FTE-type flux ropes takes place through secondary reconnection at the point of contact between the two flux ropes. The BepiColombo measurements indicate a large normalized guide field and a reconnection rate comparable to that measured at the magnetopause (~ 0.1).

## 1. Introduction

Flux transfer events (FTEs) are frequently observed near the outer boundaries, i.e., magnetopause, of planetary magnetospheres, including on Earth (e.g., Russell and Elphic, 1978; Saunders et al., 1984; Wang et al., 2005), Mercury (Russell and Walker, 1985; Slavin et al., 2009, 2010, 2012; Imber et al., 2014; Sun et al., 2020a; Zhong et al., 2020), Saturn (Jasinski et al., 2016, 2021) and Jupiter (Walker and Russell, 1985; Lai et al., 2012). Some of the FTEs have magnetic flux ropes at their cores, which consist of helical magnetic field lines surrounding stronger magnetic fields paralleling their central axes (Paschmann et al., 1982; Lee et al., 1993). These FTE-type flux ropes are created by multiple X-line reconnections in the magnetopause during intervals of significant magnetic shear across the magnetopause current sheet (Lee and Fu, 1985; Raeder, 2006). As a result, the FTE-type flux ropes signal

not only the occurrence of magnetic reconnection but their directions of travel can be used to infer the relative location of the reconnection X-lines at the magnetopause.

The FTEs usually include magnetic field lines with one end connecting to the IMF and the other to magnetospheric cusp. They transport magnetic flux from the dayside to the nightside magnetosphere that drives the Dungey cycle in planetary magnetospheres with global intrinsic magnetic fields. Sun et al. (2022) has recently reviewed the

contributions of FTE-type flux ropes to the Dungey cycle in dipolar planetary magnetospheres. In Mercury's magnetosphere, FTE-type flux ropes transport the majority (>60%) of the circulated flux (Slavin et al., 2010; Sun et al., 2020a). In contrast, FTE-type flux ropes are estimated to transport only a small portion (<5%) of the circulated flux at Earth (Lockwood et al., 1995; Fear et al., 2017). For the giant outer planetary magnetospheres at Jupiter and Saturn, they appear to transport a negligible magnetic flux (< 1%) for the solar wind-driven portion of their internal

convection (Jasinski et al., 2021).

The FTEs on Earth's magnetosphere appear most frequently during periods of the southward interplanetary magnetic field (IMF) when the magnetic shear angle across the magnetopause is larger than 90° (e.g., Rijnbeek et al., 1984; Kuo et al., 1995; Wang et al., 2006). The locations of magnetopause X-lines are closely related to the orientation of the IMF. For example, during the purely southward IMF, reconnection most likely occurs on the

magnetopause near the subsolar point (Dungey, 1961). During the purely northward IMF, reconnections occur on the magnetopause tailward of the cusp (Dungey, 1961; Song and Russell, 1992; Shi et al., 2009, 2013; Gou et al., 2016). Magnetic reconnection is also thought to occur at the dayside magnetopause under the strong radial IMF ($B_x$ dominate) (Belenkaya, 1998; Luhmann et al., 1984; Pi et al., 2017; Tang et al., 2013; Toledo-Redondo et al., 2021), but the strong radial IMF conditions are less well studied.

Coalescence, which refers to the merging of neighboring flux ropes, is thought to be an important process in space plasma physics (Biskamp and Welter, 1980; Dorelli and Bhattacharjee, 2009; Fermo et al., 2011; Hoilijoki et al., 2017). The merging of flux ropes is associated with secondary reconnection, and changes in magnetic field configuration caused by this secondary reconnection can energize particles, especially electrons (Drake et al., 2006). Furthermore, several studies have suggested that FTE-type flux ropes are initially formed between electron to ion

scales. They then grow through coalescence, thereby, increasing their magnetic flux contents and scales (Fermo et al., 2011; Akhavan-Tafti et al., 2018). NASA's Magnetospheric Multiscale (MMS) Mission (Burch et al., 2016) has provided several observations of secondary reconnections between neighboring flux ropes (see, Zhou et al., 2017), between flux rope and Earth's dipole magnetic field (Poh et al., 2019), and between interlinked flux tubes (Øieroset et al., 2016; Kacem et al., 2018).

This study investigates FTE-type flux ropes and reconnection at Earth's dayside magnetopause during BepiColombo's flyby on 10 April 2020. The paper is arranged as follows. Section 2 introduces the BepiColombo mission and the measurements during Earth's dayside magnetopause crossing. Section 3 analyzes the distribution of magnetopause reconnection with a strong radial IMF component, and the properties of the flux ropes, including a coalescence event. Section 4 provides a summary of our results.

    **2. BepiColombo Dayside Magnetopause Crossing**

**2.1.    Spacecraft and Instrumentation**

BepiColombo is a joint mission by European Space Agency (ESA) and Japan Aerospace Exploration Agency (JAXA), which consists of two spacecraft named the Mercury Planetary Orbiter (MPO) and Mercury Magnetospheric Orbiter (MMO, or Mio). These spacecraft together aim to carry out detailed investigations of

Mercury's interior, surface, exosphere, and magnetosphere (Milillo et al., 2020; Murakami et al., 2020; Benkhoff et al., 2010). The mission made its first planetary flyby maneuver at Earth on 10 April 2020 (Mangano et al., 2021), during which several instruments collected measurements. The MPO and the MMO were attached during the Earth flyby, and therefore, their measurements could be deemed as one observation point. The two spacecraft will be separated when they are scheduled to insert into Mercury's orbit by late 2025 or early 2026.

This study uses measurements collected by the magnetometer (MAG) onboard MPO (Heyner et al., 2021), and the low energy electron by Mercury Electron Analyzer (MEA) (Sauvaud et al., 2010), which is part of the Mercury Plasma Particle Experiment (MPPE) onboard MMO (Saito et al., 2021). The MPO/MAG includes one outboard sensor and one inboard sensor, and it has a sampling rate of 128 Hz. Mio/MEA has a sampling rate of 4 s. The IMF and solar wind conditions are obtained from the OMNI dataset (King and Papitashvili, 2005), which has a time

resolution of 1 minute.

**2.2.    Overview of Magnetosheath and Magnetopause**

Figure 1 shows an overview of the dayside magnetopause crossing during BepiColombo's Earth flyby. BepiColombo traveled from the magnetosheath into the dayside magnetosphere. It crossed the magnetopause at a distance of ~ 4.8 $R_E$ ($R_E$ is one Earth radius) dawnward from the subsolar magnetopause, which corresponded to a

position of (11.2, -4.8, -0.3) $R_E$ in the Geocentric Solar Magnetospheric (GSM) coordinate. During the 30 minutes interval around the magnetopause crossing (~00:05 to 00:35 UT) analyzed here, the IMF was southward with a strong radial component, i.e., the $B_x$ was the dominant component ($B_x/B_t > 0.7$ in Figure 1h). The average electron density in the magnetosheath was estimated to be ~ 10 $cm^{-3}$ based on the onboard-calculated partial moment from Mio/MEA between 00:05 and 00:28 UT. The magnetosheath plasma $\beta$ was high with a value of ~ 8.0, which was

the ratio of the thermal pressure to the magnetic pressure. The thermal pressure in the magnetosheath was calculated by assuming that the pressure balance existed across the dayside magnetopause and the thermal pressure inside the dayside magnetosphere was negligible compared to the magnetic pressure.

**3.    Magnetopause Reconnections and FTE-type Flux Ropes**

**3.1.    Identification of FTE-type Flux Ropes**

The FTE-type flux ropes were identified after the measured magnetic field was rotated into boundary normal coordinates (the LMN coordinate). The minimum variance analysis (MVA) (Sonnerup and Cahill Jr., 1967; Sonnerup and Scheible, 1998) was performed on the magnetic field measurements across the magnetopause current sheet from 00:32:30 to 00:33:25 UT to obtain the LMN coordinate. The MVA results produced L =

[0.10, 0.24, 0.97] (maximum variance direction), M = [0.12, 0.96, -0.25] (intermediate variance direction), N = [0.99, -0.14, -0.06] (minimum variance direction), and the eigenvalue ratios were $\lambda_{max}/\lambda_{int} \sim 54.3$, $\lambda_{int}/\lambda_{min} \sim 3.9$. The $\lambda_{max}$, $\lambda_{int}$ and $\lambda_{min}$ are the maximum, intermediate, and minimum eigenvalues. Both of the ratios were larger than 3 indicating that the LMN coordinate of the magnetopause was well determined [*Sonnerup & Scheible*, 1998]. The FTE-type flux ropes were identified with bipolar signatures in the normal magnetic field ($B_N$) and clear magnetic field rotation (Russell and Elphic, 1978). The identification of flux ropes also required the signature of a strong magnetic field along their central axis, i.e. the intermediate variance direction (see Figure 2 for an example, and e.g. *Slavin et al.* [2009]; *Akhavan-Tafti et al.* [2018]). Six FTE-type flux ropes were identified in this manner in the magnetosheath just upstream of the dayside magnetopause and marked with green arrows in Figure 1e and listed in Table 1.

The first possible FTE-type flux rope shown in Figure 2 was centered at ~ 00:11:04 UT when the IMF clock angle was ~ 210°, and $B_x/B_t$ was ~ 0.75. This flux rope traveled southward as inferred from the polarities of the $B_N$ variation (negative to positive, Figure 2c). The flux rope corresponded to clear enhancement in $B_M$ (Figure 2b) and field rotation in the plane of $B_{max}$-$B_{int}$ (Figure 2e). However, the enhancement in the $B_t$ strength preceded the reversal in $B_N$ could indicate that the magnetic flux was piled up or this structure was a magnetosheath structure other than a flux rope. About 2 minutes later, the clock angle increased to ~ 260°. This IMF orientation persisted for about 12 minutes, during which no FTE-type flux ropes were observed. At ~ 00:26:06 UT, the clock angle decreased from ~ 260° to ~ 210° while the ratio of $B_x/B_t$ increased to ~ 0.90. At this point, 5 FTE-type flux ropes successively appeared up to the point where the magnetopause was crossed. The travelling direction for these 5 flux ropes was inferred to be northward, again based on the $B_N$ variations. The first flux rope traveled southward indicating that the primary magnetopause X-line was initially located northward of the spacecraft. Later, the northward motion of the 5 flux ropes indicated that the primary magnetopause X-line(s) had shifted southward.

### 3.2. Reconnection X-lines from Maximum Magnetic Shear Model

To further investigate reconnection during BepiColombo's dayside magnetopause traversal, the maximum magnetic shear model (Trattner et al., 2007, 2017) was employed to deduce the location of reconnection X-lines. The magnetic shear angle plots during the intervals centered at 00:09, 00:20, 00:28 UT are shown in Figure 3. Figures 3a and 3b correspond to a distorted feature of the anti-parallel reconnection region, which has recently been termed a "Knee" event (Trattner et al., 2021). The bent shape of the anti-parallel reconnection region is associated with the field line draping in the magnetosheath during the dominant $B_x$ (significantly sunward) component in this period. Figure 3c did not provide the predicted X-line. This was because a continuous X-line along the maximum magnetic shear location was difficult to obtain under the situation of a $B_x/B_t \geq 0.9$, which was due to the lack of comprehensive study on how the significant radial IMF draping around the magnetopause influences magnetic reconnection.

In Figure 3a, BepiColombo was located southward of the predicted X-line. From Figure 3a to Figure 3b, the predicted X-line crossed the location of BepiColombo and was then located to the south of BepiColombo. The

changes in X-line locations from Figures 3a to 3b were due to the IMF clock angle decreasing around 10° together with the $B_x/B_t$ increasing from 0.78 to 0.86.

The travelling directions for the FTE-type flux ropes were consistent with the predicted locations of the reconnection X-line by the maximum magnetic shear model during the changing solar wind conditions for this magnetopause encounter. Figure 3a corresponded to the only southward traveling FTE-type flux rope, while the other five northward traveling FTE-type flux ropes were observed during the conditions shown in Figures 3b and 3c. It needs to note that the FTE-type flux ropes and reconnection exhausts should correspond to strong lateral motion as the predicted X-lines were significantly along the north-south direction. The reconnection exhausts would correspond to a strong duskward component when the spacecraft was located southward of the X-line and a strong dawnward component when it was northward of the X-line. Although the maximum magnetic shear model faces challenges in determining the draping magnetic field lines in the magnetosheath during the intervals of the dominant $B_x$ component (Trattner et al., 2007, 2012), the model predictions are consistent with our observations during BepiColombo's crossing.

### 3.3. FTE-type Flux Rope Modeling

This study employed a force-free flux rope model (Kivelson and Khurana, 1995) to fit the FTE-type flux ropes. This flux rope model starts from the periodic pinch solution (Schindler et al., 1973) of Ampere's law ($\nabla \times \vec{B} = \mu_0 \vec{J}$), where $\vec{B}$ is the magnetic field vector, $\vec{J}$ is the current density vector, and $\mu_0$ is the magnetic permeability in vacuum. Kivelson and Khurana (1995) further include the axial magnetic field component ($B_{int}$) in the periodic pinch solution. The flux rope model introduced by Kivelson and Khurana (1995) does not consider the gradient of the magnetic field along the axis of the flux rope. The self-consistent solution of the flux rope model is

$$
\begin{cases}
B_{max} = \left(\dfrac{B_T}{\chi}\right)\sqrt{1+\epsilon^2}\sinh\left(\dfrac{x_{min}}{T}\right) \\[2mm]
B_{int} = \left(\dfrac{B_T}{\chi}\right)\sqrt{1+\left(\dfrac{B_{int0}\chi}{B_T}\right)^2} \\[2mm]
B_{min} = \left(\dfrac{B_T}{\chi}\right)\epsilon\sin\left(\dfrac{x_{max}}{T}\right)
\end{cases}
\tag{1}
$$

In the equation, the $x_{min}$ and $x_{max}$ are the positions in the flux rope along the directions of $\vec{n}_{min}$ and $\vec{n}_{max}$. The $\vec{n}_{min}$, $\vec{n}_{int}$ and $\vec{n}_{max}$ refer to the local coordinate of the flux rope, which are determined from the MVA on the flux rope. The $T$ is the vertical scale of the flux rope in the $\vec{n}_{max}$ direction and the $B_T$ is the magnetic field intensity near the boundary of the flux rope along the $\vec{n}_{min}$ direction. The $B_{int0}$ is the $B_{int}$ in the background. The $\chi$ is

$$
\chi = \epsilon\cos\left(\frac{x_{max}}{T}\right) + \sqrt{1+\epsilon^2}\cosh\left(\frac{x_{min}}{T}\right)
\tag{2}
$$

In this equation, the parameter $\varepsilon$ is associated with the shape of the flux rope, i.e., from flattened to circular profiles. The axial flux content ($\Phi_{axial}$) is calculated by integrating the axial field ($B_{int}$) over the entire flux rope area,

$$
\Phi_{axial} = \int B_{int} dS
\tag{3}
$$

During the fitting, we assume that the traveling speed of flux ropes was 100 km/s, which corresponds to the average Alfvén speed in the subsolar magnetosheath. The traveling speed is required in calculating the scales and magnetic flux content for the flux ropes. The least-squares of the minimization of the magnetic field differences ($X^2$) is employed to define the best fit, which is calculated from

$$X^2 = \frac{\sum_{i=1}^{N_{point}} \sum_{j}^{max,int,min} \left[\left(B_j(i) - B_j'(i)\right)/B_t(i)\right]^2}{N_{point}} \tag{4}$$

where $B_{max}$, $B_{int}$, $B_{min}$, and $B_t$ are the components and magnitude of the measured magnetic fields and $B_{max}'$, $B_{int}'$, and $B_{min}'$ are the components of the magnetic fields from the model. The $N_{point}$ is the number of data points. We set up a threshold of $X^2 < 0.1$ to be the successful modeling.

Different from the circular profile of flux ropes resulting from the Lundquist force-free flux rope model (Lundquist, 1950; Burlaga, 1988; Lepping et al., 1990), this force-free model can result in either flattened or circular profiles of flux ropes. We use the semi-minor and semi-major to refer to the flattened features. The semi-major corresponds to the scale of flux rope along the $\vec{n}_{min}$ direction, which is close to the L direction of the magnetopause. The semi-minor correspond to the scale of flux rope along the $\vec{n}_{max}$ direction, which is close to the N direction of the magnetopause. This flux rope model is successfully applied for the flux ropes in Earth's plasma sheet (Kivelson and Khurana, 1995), on Earth's magnetopause (Zhang et al., 2008), and in Mercury's plasma sheet (Zhao et al., 2019).

Out of the 6 FTE-type flux ropes, 4 were successfully modeled. As an example, the modeling curves of the flux rope centered at 00:28:13 UT are shown in Figures 4a to 4d. In the figures, the dashed lines overlapping with the solid measured magnetic fields represent the modeling curves from the flux rope model. It can be seen clearly that the two curves are close to each other and this flux rope is well fitted by the model. The modeling results for the 4 flux ropes are summarized in Table 1. The plasma density of ~ 10 cm$^{-3}$ in the magnetosheath corresponds to an ion inertial length ($d_i$) of ~ 70 km. The two FTE-type flux ropes at 00:26:06 UT and 00:26:26 UT are in the scales of several $d_i$. The magnetic flux contents of these two flux ropes are small (~ 20 kWb). In addition, these two flux ropes correspond to the largest and smallest core fields. The other two FTE-type flux ropes at 00:28:13 UT and 00:30:26 UT are in the scales of more than 10 $d_i$. These two flux ropes contain much higher magnetic flux (~ 300 kWb and ~ 188 kWb). The analysis of the flux rope at ~ 00:28:13 UT corresponding to the highest magnetic flux content is shown in the next section. Moreover, the flux ropes at 00:26:06 UT, 00:26:26 UT, and 00:30:26 UT are close to circular profiles with the semi-minor slightly smaller than the semi-major. The flux rope at ~ 00:28:13 UT corresponds to the most flattened profile.

### 3.4. Coalescence Event

Figures 4a to 4d show the magnetic field measurements of the FTE-type flux rope centered at ~ 00:28:13 UT in the LMN coordinate. This FTE-type flux rope corresponds to the fifth green arrow counting from the leftside in Figure 1e. Figure 4c showed that the $B_N$ included two successive bipolar signatures, which implied that two smaller scale flux ropes were merging. Indeed, the hodogram in the $B_{max}$-$B_{int}$ plane in Figure 4f confirmed the field rotations of two flux ropes, named "FR#A" and "FR#B". Figure 4e further illustrated the merging of the two flux ropes and the trajectory of BepiColombo. The magenta arrow and shaded region in Figure 4e indicated the possible secondary

reconnection between FR#A and FR#B. This FTE-type flux rope with the highest flux content possibly resulted from the coalescence of two smaller-scale flux ropes.

In order to study how well aligned FR#A and FR#B were, we applied the MVA on FR#A from 00:28:03 to 00:28:09 and FR#B from 00:28:09 to 00:28:16, separately. The eigenvalue ratios were $\lambda_{max}/\lambda_{int}$ ~ 1.91 and $\lambda_{int}/\lambda_{min}$ ~ 21.7 for FR#A. The eigenvalue ratios were $\lambda_{max}/\lambda_{int}$ ~ 3.34 and $\lambda_{int}/\lambda_{min}$ ~ 12.6 for FR#B. The large values of $\lambda_{int}/\lambda_{min}$ indicated that the $\vec{n}_{min}$ were well determined for both flux ropes. The $\vec{n}_{min}$ was [-0.20,-0.58,-0.79] for FR#A and $\vec{n}_{min}$ was [0.23,-0.55,-0.80] for FR#B. The $\vec{n}_{min}$ were close to each other with a separation angle of 25°. The $\vec{n}_{min}$ obtained for the coalescence event was [-0.04,-0.49,-0.87], which were 12° and 17° away from the $\vec{n}_{min}$ of FR#A and FR#B, separately. The small separations of the $\vec{n}_{min}$ should indicate that FR#A and FR#B were well aligned. It needs to note that the coalescence signature was only observed in this FTE-type flux rope centered at ~ 00:28:13 UT. The successive bipolar signatures of the $B_{N}$ were not found in other 5 FTE-type flux ropes.

### 3.5.    Magnetopause Reconnection and Secondary Magnetic Reconnection

In Figure 5, the properties of the secondary current sheet in the coalescence event and the magnetopause current sheet are studied. For the secondary current sheet, the eigenvalue ratios were $\lambda_{max}/\lambda_{int}$ ~ 6.4, $\lambda_{int}/\lambda_{min}$ ~ 11.0 resulting from the MVA. Both of the eigenvalue ratios were larger than 3 indicating the local coordinate of the secondary current sheet was well established. The magnetic field measurements of the magnetopause current sheet were shown in the LMN coordinate.

In the reconnecting current sheet, the dimensionless reconnection rate can be determined from the ratio of the normal magnetic field component ($B_{normal}$) to the reconnecting magnetic field ($B_{inflow}$) in the inflow region (Sonnerup, 1974; Sonnerup et al., 1981; Fuselier and Lewis, 2011; Phan et al., 2001; Sun et al., 2020b). In the secondary current sheet (Figures 5a to 5d), the $B_{normal}$ was ~ 5 nT, which corresponded to the $B_{min}$ averaged from 00:28:08.8 to 00:28:09.6 UT. Here the average $B_{t}$ from 00:28:09.8 to 00:28:10.4 UT was taken as the $B_{inflow}$ (~ 36 nT). The dimensionless reconnection rate was ~ 0.14 if the reconnection occurred in the secondary current sheet. Meanwhile, the intensity of the guide field ($B_{int}$, Figure 5b) was ~ 32 nT across the current sheet, which was ~ 0.89 when normalized to the $B_{inflow}$. In the magnetopause current sheet, the $B_{normal}$ was 8.3 nT, which corresponded to the averaged $B_{N}$ from 00:32:56 to 00:33:05 UT (Figure 5g). The $B_{inflow}$ in the magnetosphere side adjacent to the magnetopause was ~ 46.1 nT, which corresponded to the averaged $B_{t}$ from 00:33:06 to 00:33:15 UT (Figure 5h). Thus, the dimensionless reconnection rate was calculated to be ~ 0.18. The guide field across the magnetopause was ~ 13 nT ($B_{M}$, Figure 5f), which was 0.28 normalized to the $B_{inflow}$.

However, it is needed to point out that the estimation of reconnection rate based on $B_{N}/B_{inflow}$ could be imprecise. For example, the uncertainties of the normal direction and the fluctuations in the field strength could influence the accuracy of the reconnection rates. As noted by Sonnerup and Scheible (1998) and Khrabrov and Sonnerup (1998), there were uncertainties in the eigenvectors determined by the MVA, which could be either statistical error or error due to the magnetic structure was not perfectly stationary and one-dimensional. By employing the method introduced by Khrabrov and Sonnerup (1998), we obtained an uncertainty of ~ 0.93 nT for the $B_{normal}$ of the secondary current sheet and ~ 0.04 nT for the magnetopause current sheet.

However, it was not certain that magnetic reconnection was occurring in the secondary current sheet or the magnetopause current sheet. There was no complimentary evidence for the magnetic reconnection since the measurements from BepiColombo were limited during the Earth flyby. The low energy electron measurements (Mio/MEA) were limited in the field of view and the time resolution was ~ 4 seconds. The MEA could not provide a complete distribution relative to the background magnetic field and its time resolution was much longer than the
time scale of the secondary current sheet. Therefore, the conclusions obtained about magnetic reconnection are tentative and further analysis about a similar event is needed, especially those measurements taken from the MMS.

## 4. Conclusions and Discussions

Our analysis of the subsolar magnetopause observations during BepiColombo's Earth flyby has produced several conclusions.

First, the BepiColombo's dayside magnetopause crossing took place during an interval when magnetosheath had a high plasma $\beta$ (~ 8) and the IMF had a strong radial component ($B_x/B_t > 0.7$). The traveling of the FTE-type flux rope suggests that the X-line crosses the location of BepiColombo. Although there is a possibility that the first and only southward travelling FTE-type flux rope is a magnetosheath structure, the predictions of the maximum magnetic shear model suggest that the X-line crosses the location of BepiColombo as well. The X-line motion is
associated with the rotation and the $x$ component increase of the IMF. BepiColombo crosses the magnetopause near the magnetic equator, and 10 April 2020 is close to the spring equinox, which indicates a small Earth's dipole tilt influence. These observations of the possible crossing of the X-line provide clear evidence of magnetic reconnection occurrence near the magnetic equator under a strong radial IMF.

Second, the properties of the FTE-type flux ropes are obtained by employing a force-free flux rope model
introduced by Kivelson and Khurana (1995). The FTE-type flux ropes correspond to scales ranging from several $d_i$ to around 20 $d_i$, and the FTE-type flux rope with a large scale and the highest magnetic flux content exhibits clear coalescence signatures. These observations strongly support the theories in which the FTE-type flux ropes grow in scales and magnetic flux contents through coalescence.

Third, magnetic reconnection in the coalescence event and the magnetopause current sheet is investigated. The
reconnection rate of the secondary reconnection (0.14) is comparable with the reconnection rate of dayside magnetopause (0.18). However, the secondary reconnection corresponds to a larger normalized guide field (0.89) and a magnetopause reconnection (0.28). However, there is no complimentary evidence that magnetic reconnection is occurring in the secondary current sheet and magnetopause current sheet. Therefore, the conclusions about magnetic reconnection are tentative.

The large guide field of the secondary magnetic reconnection during the coalescence observed by BepiColombo is likely a common feature. For example, Zhou et al. (2017) reported a coalescence event with a strong guide field. We suggest that these large guide fields should be included in future simulations, which investigate the particle energizations due to coalescence. The large guide fields may influence the reconnection rate as suggested by Pritchett and Coroniti (2004) and Ricci et al. (2004), and therefore affect the energization of particles during the

coalescence. Furthermore, a recent investigation also suggests that a large guide field might limit the ability of Fermi acceleration during the coalescence (Montag et al., 2017).

Finally, the FTE-type flux rope containing the coalescence signature has a scale of ~20 $d_i$. Therefore, the secondary reconnecting current sheet embedded within the FTE-type flux rope is likely with a scale smaller than 20 $d_i$. We want to note that the secondary reconnection during the coalescence of flux ropes share some similarities with the

electron-only reconnection associated with the magnetosheath turbulence, whose reconnecting current sheet has scales smaller than 10 $d_i$ and is accompanied by a large guide field as revealed by MMS measurements (Phan et al., 2018; Stawarz et al., 2019) and simulations (Califano et al., 2020). Therefore, it is likely that the secondary reconnection associated with coalescence is electron-only magnetic reconnection, which certainly deserves a detailed study.


**Data availability**

The measurements from Mio/MEA and MPO/MAG analyzed in this study are available in the supporting information. The data archiving is underway. Mio/MEA data will be able to be accessed from the AMDA science analysis system (http://amda.cdpp.eu) provided by the Centre de Données de la Physique des Plasmas (CDPP)

supported by CNRS, CNES, Observatoire de Paris, and Université Paul Sabatier Toulouse. MPO/MAG data will be available from https://archives.esac.esa.int/psa/#!Home%20View. OMNI dataset is available at https://omniweb.gsfc.nasa.gov/.


**Author contributions**

W. J. S. led the work, identified the events, conducted the data analysis of the dataset, and wrote the manuscript. W. J. S., J. A. S., and R. N. jointly designed the work. D. H. and J. Z. D. M. provided knowledge of the MPO-MAG instrument and the MPO-MAG data. S. A. and N. A. provided knowledge of the Mio-MEA instrument and the Mio-

MEA data. K. J. T. provided Figure 3 and the relevant descriptions. J. T. Z. performed force-free fittings of the flux ropes. All authors discussed and contributed to the manuscript.

**Competing interests**

The authors declare no competing interests.


**Acknowledgment**

The BepiColombo project is supported by ESA and JAXA. W. J. S. and J. A. S. were supported by NASA Grants NNX16AJ67G and 80NSSC18K1137. N.A. and S.A. acknowledge the support of CNES for the BepiColombo mission. The research at LASP (K. J. T.) is supported by NASA grants NNG04EB99C and 80NSSC20K0688. W. J.

S. thanks Dr. Gangkai Poh for helpful discussions.

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

**Figures and Tables**

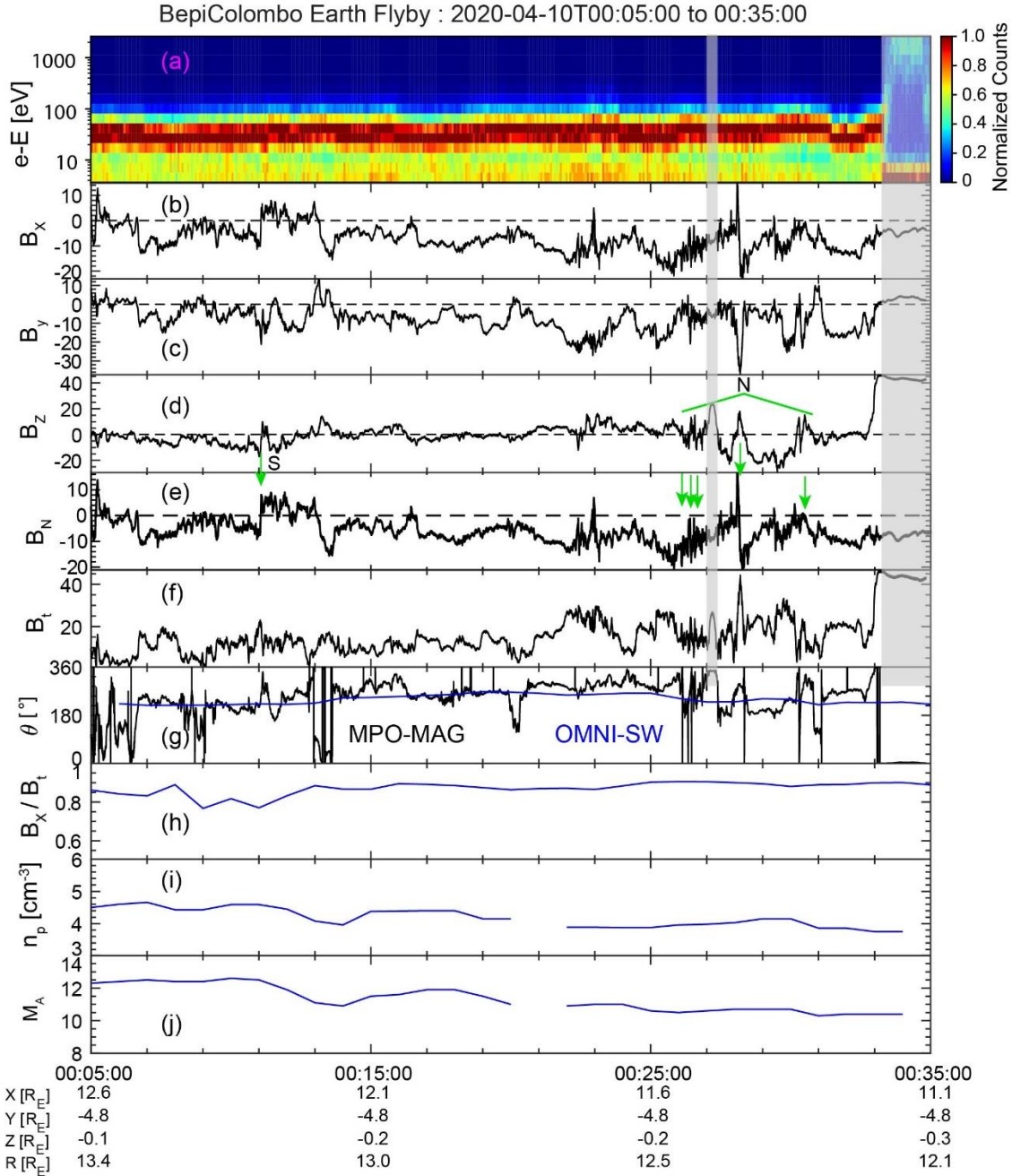

**Figure 1. The electrons and magnetic field measurements of the dayside magnetopause during BepiColombo's Earth flyby.** (a) the time-energy spectrogram of normalized electron counts from Mio/MEA, (b) magnetic $x$ component $B_x$, (c) $y$ component $B_y$, (d) $z$ component $B_z$, (e) the magnetic component normal to the magnetopause $B_N$, (f) the magnetic field intensity $B_t$, (g) the clock angle ($\theta$), (h) $B_x/B_t$ of the IMF, (i) solar wind number density ($n_p$), (j) solar wind Alfvénic Mach number ($M_A$). All quantities are in the Geocentric Solar Magnetospheric (GSM) coordinate. The green arrows in (e) indicate the six FTE-type flux ropes. "S" indicates southward traveling and "N" northward traveling. The $\theta$ in (f) is defined as $\arctan(B_y/B_z)$, ranging from 0° to 360°. In (g), the black line is from the measurements of MPO/MAG, and the blue line is from OMNI.

**Table 1. List and properties of FTE-type flux ropes observed during BepiColombo's dayside magnetopause crossing**

| # | Time | Duration (s) | Travelling Direction | Core Field Intensity (nT) | Scale (km) [b] | Flux Content (kWb) | $X^2$ |
|---|------|-------------|---------------------|--------------------------|----------------|--------------------|-------|
| 1 | 00:11:04 | ~ 12 | Southward | — [a] | — | — | — |
| 2 | 00:26:06 | ~ 7 | Northward | ~23.9 | 462, 388 (0.84) | ~13.7 | ~0.04 |
| 3 | 00:26:26 | ~ 6 | Northward | ~60.8 | 565, 524 (0.93) | ~22.5 | ~0.04 |
| 4 | 00:26:35 | ~ 4 | Northward | — | — | — | — |
| 5 | 00:28:13 | ~ 20 | Northward | ~41 | 1745, 1281 (0.73) | ~300 | ~0.08 |
| 6 | 00:30:26 | ~ 15 | Northward | ~45.2 | 1853,1745 (0.94) | ~188 | ~0.08 |

575

[a] "—" indicate that the values are not determined by the flux rope model. See the text for more information on the flux rope modeling.

[b] Scale contains semi-minor, semi-major, and the ratio between semi-minor and semi-major refers to the flattened profile. See the text for more information.

580

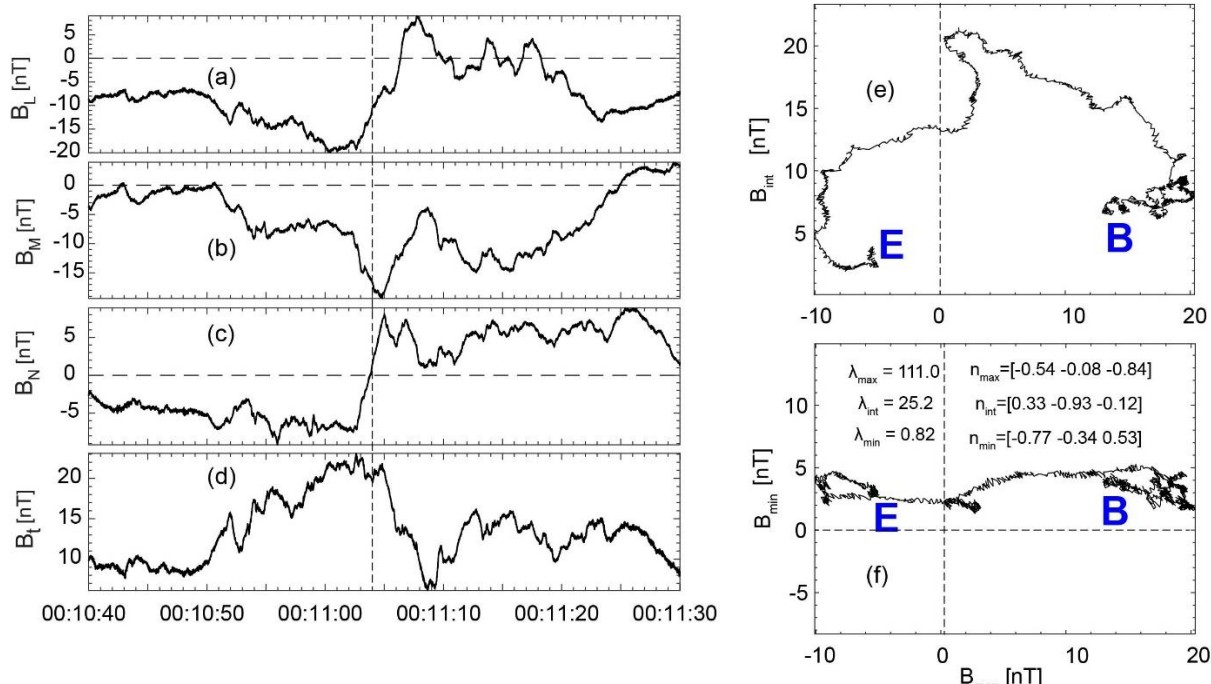

**Figure 2. The southward traveling FTE-type flux rope centered at ~ 00:11:04 UT.** (a) magnetic field component in the L direction, $B_L$, (b) magnetic field component in the M direction, $B_M$, (c) magnetic field component in the N direction, $B_N$, (d) $B_t$. This LMN is the local coordinate of the magnetopause. (e) and (f) are the hodograms of the magnetic field measurements under the local coordinate of the flux rope. The "B" and "E" indicate the beginning and the end of the data points.

**Figure 3. Magnetic shear angle plots on the magnetopause surface during BepiColombo's dayside magnetopause crossing, which are obtained through the maximum magnetic shear model** [*Trattner et al.*, **2007**]. (a), (b), (c) correspond to the IMF averaged from 00:05 to 00:13 UT, 00:16 to 00:24 UT and 00:24 to 00:33 UT, respectively. The black circle represents the terminator plane separating the dayside magnetopause from the tailward magnetopause. The grey line represents the predicted magnetopause reconnection line. White areas correspond to the magnetic shear angle is within 3° of 180°. The black dots are the location of BepiColombo ("BC"). In (c), the predicted X-line is not provided. This is because a continuous X-line along the maximum magnetic shear location is difficult to obtain under the situation of a $B_x/B_t \geq 0.9$,

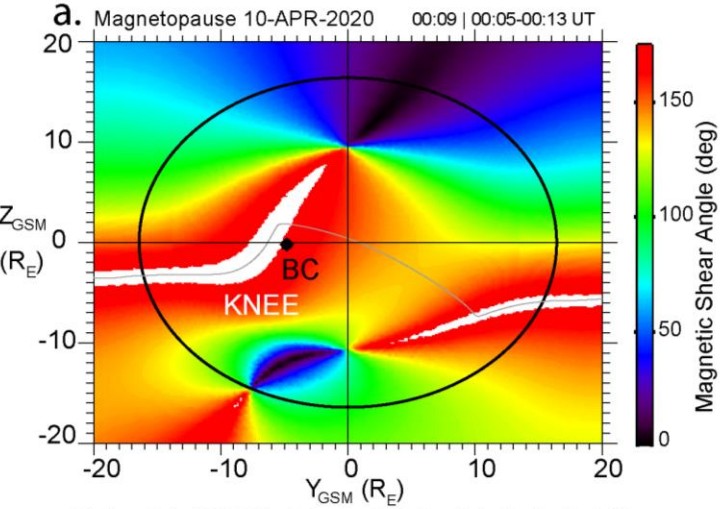

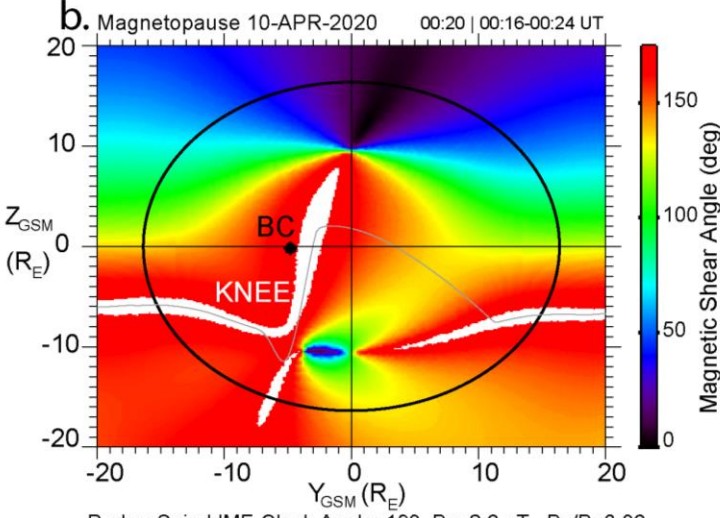

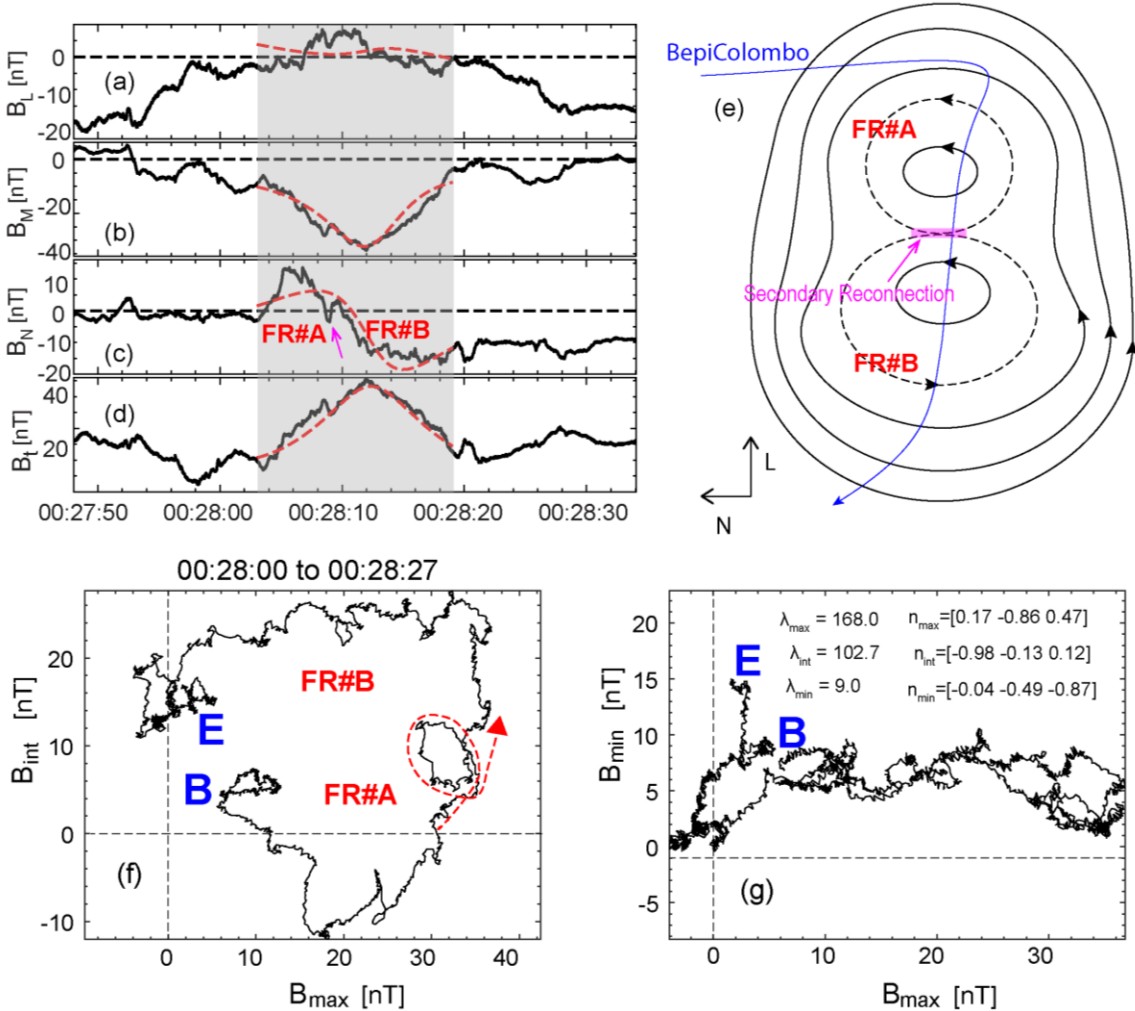

**Figure 4. Overview of the flux rope centered at ~ 00:28:13 UT with the coalescence feature.** (a) $B_L$, (b) $B_M$, (c) $B_N$, (d) $B_t$. The dashed lines are obtained from the flux rope model. This LMN is the local coordinate of the magnetopause. See the text for more information. (e) An illustration of the coalescence event and the BepiColombo's trajectory. The secondary reconnection site is marked by the magenta region. (f) and (g) are the hodograms of the magnetic field measurements under the local coordinate of the flux rope. The "B" and "E" indicate the beginning and the end of the data points. FR#A and FR#B indicate two flux ropes.

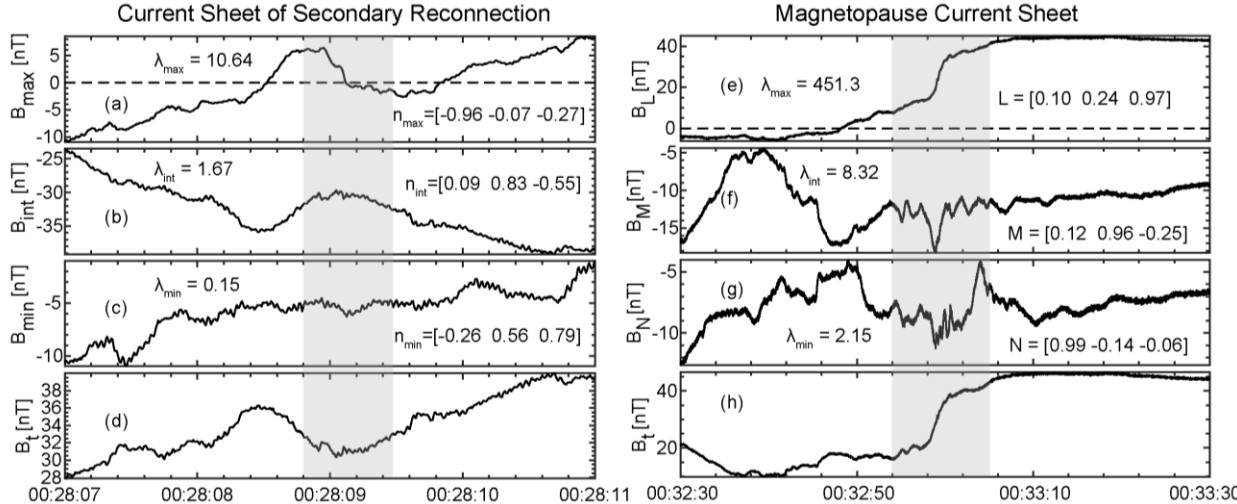

**Figure 5. The magnetic field measurements under their separately local coordinate for the reconnecting current sheet of the coalescence event and the magnetopause current sheet.** (a) to (d) are for the secondary current sheet of the coalescence event. The $\vec{n}_{min}$, $\vec{n}_{int}$ and $\vec{n}_{max}$ refer to the local coordinate of the secondary current sheet. (e) to (h) are for the magnetopause current sheet. The LMN local coordinate of the magnetopause current sheet is used. The eigenvalues and corresponding eigenvectors result from the MVA.