# Peer review of "Dayside magnetopause reconnection and flux transfer events: BepiColombo Earth-flyby observations"

_Annales Geophysicae, 2021_

## Author Comment (AC2)

We would like to thank the reviewer for his/her help to review our paper. The comments and suggestions are encouraging and useful in revising the manuscript. We have responded to the reviewer's comments below.

**Summary**

The manuscript reports experimental observations made by the BepiColombo mission during an Earth flyby on 10 April 2020. It focusses on the crossing of the Earth's magnetopause, and uses data from the MPO magnetometer and MMO electron spectrometer. A series of flux ropes are identified in the data, and their qualitative motion is compared with models of the expected X-line location, showing good agreement. A flux rope model is also used to calculate various properties. Evidence is presented showing that one flux rope in facts consists of two discrete structures separated by a thin current sheet, and it is argued that coalescence via magnetic reconnection is occurring.

Overall the manuscript makes a new and important contribution to our knowledge and understanding of flux ropes on the Earth's magnetopause and subject to minor corrections I would recommend publication. This is based on a need to clarify certain aspects of the analysis, and also soften some of the statements, particularly on the existence or otherwise of active reconnection at the thin current sheet seen in the proposed coalescence structure. The comments below expand on this in more detail, and are presented in the order they appear in the manuscript. In addition I would also recommend a careful proof-reading of the manuscript to address some mistakes in English and grammar which I have not listed.

**Detailed comments**

Line 35. A caveat to this paragraph is that the transport of flux means they are closed to the ionosphere at one end.

We agree. We have added a sentence here to emphasize this point.

"The FTEs usually include magnetic field lines with one end connecting to the solar wind and the other to the cusp. They contribute to the transport of magnetic flux from the dayside to the nightside magnetosphere that drives the Dungey cycle in dipolar planetary magnetospheres."

**Line 49. It would be worth citing recent studies on this topic by e.g. Toledo Redondo et al. https://doi.org/10.1029/2021JA029506**

We have cited this newly published paper.

Line 57. I am not sure it is not quite right to say that e.g. Oieroset et al. and Kacem et al. observed coalescence of flux ropes. There is so-called secondary reconnection, but the analysis suggests it is of a type where interlinked flux tubes are in tension against each other and then reconnecting.

The reviewer is correct. We have rewritten this place to include some detail information.

"NASA's Magnetospheric Multiscale (MMS) Mission (Burch et al., 2016) has provided several observations of secondary reconnections between neighboring flux ropes (see, Zhou et al., 2017), between the flux rope and Earth's dipole magnetic field (Poh et al., 2019), and between interlinked flux tubes (Øieroset et al., 2016; Kacem et al., 2018)."

**Line 70. Add a comment to mention that MMO and MPO are attached and so for the purposes of the flyby there is one spacecraft/observation point.**

Thanks for the suggestion. We have added "The MPO and the MMO were attached during the Earth flyby. Therefore, their measurements could be deemed as one observation point during the Earth flyby."

**Line 80. Maybe mention the local time here, or the exact location in GSE**

We have added the exact location in this place, which is (11.2, -4.8, -0.3)  $R_E$  in the GSM coordinate.

Line 101. In the overview figure it may help to plot the magnetic field in the boundary normal coordinate system as well. This would help show the existence of the flux ropes even more clearly, and allows the reader to see the polarity in B\_N (negative/positive for southward and positive/negative for northward) more clearly.

In Figure 1, we have added the magnetic field component normal to the magnetopause  $B_N$  (Figure 1d).

Line 126. The location of the model X-line in Figure 2a is nicely consistent with the observations, but I found Figure 2b and 2c harder to understand. In Figure 2c there is no predicted X-line? In Figure 2b it is highly tilted, and so while I agree that it would lead to northward motion in the general sense, there would also presumably be lateral motion with the reconnection exhausts pointing northward and dawnward. This should be clarified.

Figure 2c corresponds to a very strong IMF  $B_x$  ( $B_x/B > 0.90$ ). Under this situation, a continuous X-line along the maximum magnetic shear location is very difficult to draw. Since IMF field line would drape over the magnetopause under such a large Bx, and we are still missing a comprehensive study on how the IMF draping works. Therefore, we did not draw a predicted X-line for Figure 2c.

However, as outlined in Trattner et al. [2007], large IMF Bx cases seem to settle for a magnetic reconnection location in the antiparallel reconnection sites (white areas in Figure 2). The reconnection site in Figure 2c is therefore predicted to be in the white areas of the magnetopause magnetic shear angle plot which is – with respect to the satellite location at the magnetopause and associated observations - similar to the location outlined in Figure 2b.

This has been clarified in the revised manuscript.

Yes. We agree with the lateral motion of the reconnection exhaust. We have clarified this in the revised manuscript.

Line 142. Is it necessary to assume a specific speed for the fit, or just that it is moving at constant speed? I could understand that the speed is necessary to get the estimate of flux content.

The fitting does not require a background speed. To obtain the scale and flux content, we need a background speed for the FTE-type flux ropes.

This has been clarified in the revised manuscript.

Line 149. The model gives interesting extra information about the flattening of the flux ropes. Apologies if I missed this, but can you add some text to discuss how the flux ropes are flattened, is it in the direction of motion, or along the normal to the current sheet. Also is the flattening significant? Is there much deviation from a circular profile. I think these points would be of interest to other readers studying this problem. We have added further explanations of the flatten profile of the flux ropes. "The semi-major corresponds to the scale of flux rope along with  $\vec{n}_{min}$ , which is close to L direction of the magnetopause. The semi-minor correspond to the scale of flux rope along with  $\vec{n}_{max}$ , which is close to the N direction of the magnetopause."

In Table 1, the column of scale has been added a ratio of semi-minor to semi-major. In the text, we have added the following descriptions.

"the flux ropes centered at 00:26:06 UT, 00:26:26 UT, and 00:30:26 UT, are close to circular profiles with the semi-minor slightly smaller than the semi-major. The flux rope centered at  $\sim$  00:28:13 UT includes the strongest flatten profile."

**Line 166. This is a very nice observation. For completeness, are you able to model FR#A and FR#B independently – is it possible to say anything about how well aligned they are?**

We have tried to model FR#A and FR#B separately, but we did not obtain reasonable results. Therefore, we have further analyzed the results from the MVA technique. The applications of MVA on FR#A and FR#B.

FR#A (00:28:03 to 00:28:09),  $\lambda_{max}/\lambda_{int} \sim 1.91$ ,  $\lambda_{int}/\lambda_{min} \sim 21.7$ ,  $\vec{n}_{min} = [-0.20, -0.58, -0.79]$

FR#B (00:28:09 to 00:28:16),  $\lambda_{max}/\lambda_{int} \sim 3.34$ ,  $\lambda_{int}/\lambda_{min} \sim 12.6$ ,  $\vec{n}_{min} = [0.23, -0.55, -0.80]$

Only the  $\vec{n}_{min}$  are well determined in both cases, which are close to each other with a separation angle of 25°.

The  $\vec{n}_{min}$  obtained for the whole coalescence event is [-0.04,-0.49,-0.87], which are 12° and 17° away from the  $\vec{n}_{min}$  of FR#A and FR#B separately.

We believe that the small separations of the  $\vec{n}_{min}$  between the FR#A and FR#B shall indicate they are well aligned. We have added some explanations of these results in the revised manuscript.

Line 178. The existence of the thin current sheet separating the two flux ropes is clear and therefore a site where reconnection can occur. But I think the statements should be softened a bit here, because on the basis of the data alone, it is not 100% certain that reconnection is occurring, as there is no complementary evidence. I know that jets cannot be observed, but is there any evidence for e.g. Hall magnetic field signatures or other structure that would point to active reconnection ongoing? With a guide field of 0.28 this signature would be somewhat distorted but should be visible, and could be related to the more-negative deviation in B\_M

where B\_L reverses. In the electron data is there any evidence for localised heating etc. (although this would likely be a weak signature and maybe difficult to observe in the 4s cadence data)? Also is it possible to use the electron data to understand the connectivity of the plasma through the whole observation of FR#A and FR#B.

We agree with point. Yes. It is hard to certain that reconnection is occurring without complementary evidence.

We did not see clear Hall magnetic field signatures even with a large guide field. The  $B_{int}$  in Figure 4b did not become more negative but less negative when  $B_{max}$  reversed. We think that this could be due to either the spacecraft crossed the center of reconnection X-line or the reconnection did not occur.

In this figure, we have provided the low-energy electron measurements associated with the coalescence event. The shaded region include the flux ropes.

During cruise phase including planetary flybys, the Mio spacecraft needs to be shielded by its Sunshield (MOSIF), and so the measurements of Mio/MEA are limited in field of view. We cannot obtain a complete distribution of electrons relative to the background magnetic field. Therefore, it is difficult to obtain further information from the low energy electron measurements as well.

In conclusion, we agree with the reviewer to soften our conclusion about the secondary magnetic reconnection. We have added more discussions in the manuscript.

---

## Author Comment (AC3)

We would like to thank the reviewer for his/her help to review our paper. The comments and suggestions are encouraging and useful in revising the manuscript. We have responded to the reviewer's comments below.

This paper presents some first observations from BepiColombo, taken as it crossed the terrestrial magnetopause during its Earth flyby. The authors present observations of six flux transfer events (FTEs). They conclude that Bepi crossed the magnetopause near the dayside reconnection line, which moved such that a combination of northward-travelling and southward-travelling FTE flux ropes were observed. They also infer the occurrence of a coalescence event whereby two smaller flux ropes merge, supporting theoretical arguments that FTE flux ropes can grow in this manner.

The paper is succinct and generally clear; if substantiated, the conclusions make a contribution to the body of knowledge, and it is nice to see some early results from BepiColombo. I am not aware of northward and southward-moving flux ropes having been observed on the same magnetopause crossing, and the idea of flux rope coalescence is relatively new. I just have two concerns on which I would appreciate it if the authors can provide further reassurance or clarification:

1) As plotted in Figure 1, these are not the clearest examples of flux transfer events, though this could be due to the temporal scale of the plot (relative to the scale of the signatures) and the fact that the data are plotted in GSM, rather than boundary normal coordinates. [Only one event is shown in boundary normal coordinates, in Figure 3.] Although the normal direction quoted on line 94 is predominantly along the GSM X direction, small differences can obscure signatures, and the fact that Bx seems to alternate between periods of about +5 nT and about -10 nT does suggest that Xgsm does not approximate the local magnetopause normal particularly well. For example, the signature for the first event seems to be more of a step function from a generally-negative to a generally-positive Bx orientation, and (as far as I can make out from the figure), the enhancement in magnetic field magnitude precedes the Bn/Bx reversal, rather than being centered on it (as would be expected for a flux rope). This does inject some doubt into the identification of this event, and that in turn undermines the conclusion of the X-line moving across Bepi. The magnetosheath field is highly structured, and a skeptic might wonder if this first signature is simply associated with some rotation of the magnetosheath field, rather than the passage of a flux rope. The remaining events are too small and close together to be able to see confidently in Figure 1, with the exception of the penultimate event. Can I suggest the authors plot all six events in a boundary normal coordinate frame, to improve confidence that these events are reliably identified? Hopefully this will rebut any skepticism, but if the signatures are as unclear in a boundary normal coordinate system, it might be worth considering whether there is any further evidence in

**support of the first event, particularly, being an FTE (or reducing the strength of the claim about the motion of the X-line past Bepi).**

In the revised Figure 1, a panel of  $B_N$  has been added. We have added a new Figure 2 to include the only southward traveling FTE-type flux rope. A shorter interval of the only southward traveling FTE-type flux rope and the hodograms of the magnetic field measurements are included in Figure 2.

We could not completely exclude the possibility that this structure was a magnetosheath structure. But based on what we observed, we would conclude that this is a southward traveling FTE-type flux rope.

Figure 2. The southward traveling FTE-type flux rope centered at ~ 00:11:04 UT. (a)  $B_L$ , (b)  $B_M$ , (c)  $B_N$ , (d)  $B_t$ . This LMN is the local coordinate of the magnetopause. (e) and (f) are the hodograms of the magnetic field measurements under the local coordinate of the flux rope. The "B" and "E" indicate the beginning and the end of the data points.

2) The authors have shown evidence of a magnetic shear, and hence current sheet, but as I understand it, evidence of active reconnection relies on the ratio of the normal/tangential field components. The authors note that uncertainties in the normal direction will influence this (line 185) - is there any other evidence that the authors can present to support this conclusion? (Or can they quantify the uncertainty on the minimum variance direction and how that translates into uncertainty on the magnetic field component normal to the current sheet?)

We have employed the minimum or maximum variance analysis (MVA) to obtain the normal direction of the current sheet. As noted by Sonnerup & Scheible (1998), the MVA requires the magnetic structure to be stationary and one-dimensional. However, in situ measured structures are hardly 100% stationary and one-dimensional. Therefore, there shall be uncertainties of the normal direction determined by MVA.

There are several studies introducing how to estimate the uncertainties of the orientations of the eigenvectors. See, section 8.3 in Sonnerup, B. U. Ö., & Scheible, M. (1998). Minimum and maximum variance analysis. In G. Paschmann & P. W. Daly (Eds.), Analysis methods for multi-spacecraft data (pp. 185-220). Noordwijk, Netherlands.: ESA Publication.

As an example, we applied the method introduced by Khrabrov and Sonnerup [1998]. The delta  $B_N$  for the secondary current sheet can be determined as ~ 0.93 nT, and for the magnetopause current sheet is ~ 0.04 nT. Those values although small, but can have some influence on the dimensionless reconnection rate.

This has been discussed in the revised manuscript.

Aside from the above, I had a few minor comments:

Line 18: "flux rope" -> "flux ropes"

Done.

Lines 45 and 46: "reconnections occur" -> "reconnection occurs"

Done.

Line 71: "and" missing after Heyner reference

Done.

**Line 73: outbound/inbound should be outboard/inboard?**

Done.

**Line 137: xmin and xmax are the locations of what? And what is meant by ''along with nmin and nmax''?**

The xmin and xmax are the positions in the flux rope along with n min and nmax.

**Line 140: What is the physical meaning of chi?**

In the equation of chi, the parameter  $\varepsilon$  is associated with the shape of the flux rope, i.e., flatten or circular profiles.

**Line 154: I am confused by the text structuring here, as Figure 3 is introduced but you then go on to talk about flux ropes (at 00:26:06 and 00:26:26 UT) that are not shown in the figure. Should the sentence introducing Fig 3 move to the next paragraph?**

The purpose is to introduce a successful modeling example. We have rewritten this sentence.

"Out of the 6 FTE-type flux ropes, 4 were successfully modeled. As an example, the modeling curves of the flux rope centered at 00:28:13 UT are shown in Figures 4a to 4d. In the figures, the dashed lines overlapping with the solid measured magnetic fields represent the modeling curves from the flux rope model. It can be seen clearly that the two curves were close to each other and this flux rope was well fitted by the model. The modeling results for the 4 flux ropes were summarized in Table 1. The plasma density was ~ 10 cm-3 corresponding to an ion inertial length (di) of ~ 70 km. The two FTE-type flux ropes centered at 00:26:06 UT and 00:26:26 UT were in the scales of several di. The magnetic flux content of these two flux ropes was small (~ 20 kWb). In addition, these two flux ropes centered at 00:28:13 UT and 00:30:26 UT were in the scales of more than 10 di. These two flux ropes contained much higher magnetic flux (~ 300 kWb and ~ 188 kWb). The analysis of the flux rope centered at ~ 00:28:13 UT corresponding to the highest magnetic flux content is shown in the next section."

Line 161: Word missing after "next"

"in the next section"

**Line 163: There seem to be some words missing from this sentence**

Corrected. "Figure 3 shows the magnetic field measurements of the FTE-type flux rope centered at  $\sim 00:28:13$  UT in the LMN coordinate."

**Line 164: I am inferring that the two successive bipolar signaures mentioned here correspond to two green arrows in Figure 1, but please clarify.**

No. It corresponds to the fifth green arrow in Figure 1e. It has been clarified in the manuscript.

**Line 168: "was clearly resulted" - does not make sense**

It has been changed into "possibly resulted from".

---

## Author Response (AR2)

We would like to thank the reviewer for his/her help to review our paper. The comments and suggestions are encouraging and useful in revising the manuscript. We have responded to the reviewer's comments below.

*The authors have addressed my second main point, but the inclusion of Figure 2 has not done much to allay my concerns about identification of the first event, which provides the premise for the conclusion about the X-line crossing the location of BepiColombo. Whilst there is a reversal in the Bn trace, it seems to be more of a (slanted) step function rather than a bipolar signature per se, and as I suggested may be the case in my previous review, the enhancement in the total field strength precedes the reversal in Bn, rather than being centered on it - so this seems to be contrary to what one would expect for a flux rope signature? The authors state in their response that they could not completely exclude the possibility that this structure was a magnetosheath structure - I would suggest that this is a strong possibility. I therefore do suggest acknowledging this possibility in Section 3.1, and softening the conclusion accordingly.*

The conclusion that the first structure is a FTE-type flux rope has been softened. We have made changes in Section 3.1. (lines 117 to 119), Conclusions (lines 249 to 252), and Abstract (line 20).

*Line 155 - I presume that the vectors nmin, nint and nmax are determined from minimum variance analysis on the flux rope, but please state so explicitly.*

Yes. The vectors are determined from minimum variance analysis. This has been added.

*The revised paper would still benefit from careful proof-reading/copyediting - I have not listed all language issues, but there are two in particular that impact on clarity:*

We have polished the English writing. Annales Geophysicae will provide English language copy-editing. We will ask for that.

*Line 155 onwards - there are several instances where the authors talk about positions/scale/intensity etc "along with [vector]" (lines 155, 156, 157, 171, 172) - I think the authors mean "in the [vector] direction" or "along the [vector] direction".*

Thanks! In these and other places, "along with" has been corrected according to the reviewer's suggestions.

**Line 187 - by "includes the strongest flatten profile" do you mean "corresponds to the most flattened profile"?**

Yes. It has been changed.